# Migration Modeling as a Valuable Tool for Exposure Assessment and Risk Characterization of Polyethylene Terephthalate Oligomers

**DOI:** 10.3390/molecules28010173

**Published:** 2022-12-25

**Authors:** Verena N. Schreier, Alex Odermatt, Frank Welle

**Affiliations:** 1Division of Molecular and Systems Toxicology, Department of Pharmaceutical Sciences, University of Basel, 4056 Basel, Switzerland; 2Swiss Centre for Applied Human Toxicology (SCAHT), University of Basel, 4055 Basel, Switzerland; 3Product Safety and Analytics Department, Fraunhofer Institute for Process Engineering and Packaging (IVV), 85354 Freising, Germany

**Keywords:** migration modeling, food contact material, food packaging, polyethylene terephthalate, oligomer, exposure assessment, risk assessment

## Abstract

Polyethylene terephthalate (PET) is one of the most widely used food contact materials due to its excellent mechanical properties and recyclability. Migration of substances from PET and assessment of compliance are usually determined by experimental testing, which can be challenging depending on the migrants of interest. Low concentrations and missing reference standards, among other factors, have led to inadequate investigation of the migration potential of PET oligomers. Migration modeling can overcome such limitations and is therefore a suitable starting point for exposure and risk assessment. In this study, the activation energy-based (E_A_) model and the A_P_ model were used to systematically evaluate the migration potential of 52 PET oligomers for 12 different application scenarios. Modeling parameters and conditions were evaluated to investigate their impact and relevance on the assessment of realistic exposures. Obtained results were compared with safety thresholds known from the concept of toxicological thresholds of concern. This allowed the evaluation and identification of oligomers and/or applications where migration or exposure levels may be associated with a potential risk because they exceed these safety thresholds. Overall, this study demonstrated that migration modeling can be a high-throughput, fast, flexible, and suitable approach for comprehensive exposure assessment.

## 1. Introduction

Polyethylene terephthalate (PET) is currently one of the most widely used materials for food packaging in Europe [1] due to its advanced mechanical properties, clarity, low gas permeability, and relatively low costs [2,3,4]. It is utilized for the production of beverage bottles, packaging and microwave trays, blisters, and multilayer packaging films [1,4,5]. The compliance evaluation of such food contact articles is provided by specific and overall migration tests using food simulants under specific test conditions [6]. In addition to regulatory compliance, migration testing can also be used in support of exposure assessment and risk characterization for non-regulated substances such as non-intentionally added substances (NIAS) [7,8]. However, experimental migration testing is time-consuming, inflexible with respect to temperature variations and packaging system heterogeneity, and in many cases even impossible due to technical or analytical limitations, resulting in the lack of migration data for many NIAS [9,10,11,12].

PET oligomers can be considered as NIAS. The formation of polymer-specific oligomers is unavoidable and can generally be attributed to the thermal or hydrolytic degradation of polymer chains during production of food contact articles [13]. PET oligomers occur in crude and complex mixtures, at varying and often low concentrations, which require sophisticated analytical methods with low limits of detection and quantification. As a necessary prerequisite for quantification, reference standards must be available. The commercial availability of reference standards for PET oligomers, however, is very limited, impeding the experimental quantification of PET oligomers in food (simulants). In addition, the European regulation [6] includes the use of ethanolic food simulants for the compliance assessment of food contact materials (FCMs). However, PET in contact with aqueous ethanol mixtures causes swelling of the material, leading to overestimated migration of oligomers and other migrants [14,15,16]. Overestimating migration levels is beneficial for safety and regulatory compliance, but it is less useful for risk and exposure assessment as it can lead to an overestimation of the actual risk. Elevated temperatures during the migration experiments in combination with aqueous ethanol solutions may also force the formation, hydrolysis [17], or modification of oligomers, rendering the experimental conditions for realistic quantification and evaluation of migrated amounts of PET oligomers inappropriate [18]. Oligomers are usually identified through untargeted screening methods, which are useful for increasing knowledge on the identity of potential migrants [12,19,20,21]. However, these methods are less sensitive compared to targeted quantification and the analysis of complex mixtures is challenging; thus, the identification of all substances, especially those present at low concentrations, is usually not feasible. Therefore, it can be anticipated that some oligomers that migrate from PET remain un-identified.

PET oligomers are not regulated by the European packaging legislation and therefore no specific migration limits have been established, nor has any risk assessment been conducted to date. This also applies to most oligomers originating from other packaging materials. A common practice to circumvent this safety gap is to restrict the migration of oligomers (<1000 g/mol) by applying migration limits of 50 µg/kg [22,23]. Another and more general approach is the use of the toxicological threshold of concern (TTC) concept to estimate Cramer classifications for oligomers and to set migration values into a toxicological assessment context [24,25]. Linear PET oligomers were previously categorized as Cramer Class I substances with an respective exposure limit of 30 µg/kg body weight/day and cyclic oligomers as Cramer Class III substances with a lower exposure limit of 1.5 µg/kg body weight/day [26]. Considering the many hurdles mentioned above, testing migration to such thresholds is challenging, leading to knowledge gaps and uncertainties regarding consumer exposure and risk assessment of these substances. Thus, alternative approaches are urgently needed.

Significant progress has been made in migration modeling over the past two decades [10,27,28,29,30]. It was demonstrated that the prediction of migration can not only overcome experimental limitations, but is also a substantially faster and often more appropriate approach for realistic exposure assessments and low-diffusion food contact materials such as PET. Further development of methods for predicting diffusion coefficients resulted in the validation of the first parameters for diffusion modeling for PET, confirming the applicability of the methodology used in the present study [9]. These advances and the existing limitations associated with experimental testing offer migration modeling as a tool that can be considered a suitable and even more high-throughput type alternative for assessing compliance of PET packaging materials.

In order to understand the potential migration of PET oligomers under realistic conditions of use, migration modeling was employed in this study. The goal of this evaluation was to determine the potential level of consumer exposure that may be linked to specific oligomers and/or applications. Through this work, we aimed to show the usefulness of migration modeling for PET oligomers and its ability to support the compliance and risk assessment process for this group of substances.

## 2. Results and Discussion

### 2.1. PET Oligomers for Migration Modeling

For the migration modelling approach, 33 previously identified PET oligomers and 19 hypothetical molecules that are potentially present in the polymer were selected (Table 1). The reported oligomers were extracted from the FCCmigex database [31,32]. The reasoning for the structure of the hypothetical oligomers is based on the molecular weight (M_W_), reported oligomers, and the chemistry used for synthesis. PET hexamers were chosen as a threshold size for the inclusion of oligomers in this study, with the exception of the first series of oligomers, where it is the octamer. These size thresholds were chosen because larger oligomers significantly exceed a molecular weight of 1000 g/mol. Such substances are associated with a much lower intestinal absorption potential after exposure, resulting in a reduced risk to the consumer [6]. In the polymerization process of PET, terephthalic acid (TPA) or terephthalic acid dimethyl ester are used as main monomers together with ethylene glycol (EG) [33]. To adapt the properties of the PET packaging materials, isophthalic acid (IPA) can be used as a co-monomer. Accordingly, PET oligomers are substances built up from TPA and EG, may be linear or cyclic, and each TPA may be replaced by an IPA unit. Additional EG units such as diethylene glycol (DEG) can also be introduced into the oligomers as a side reaction, leading to a complex situation due to the high number of possible additional isomers. However, the migration and also its prediction are the same or comparable for all isomers, since the identical molecular weight (M_W_) or nearly identical molecular volumes (M_V_) are used for migration modeling for such substances.

The selection of oligomers used in this study is represented in Table 1 and individually listed substances represent a group of isomers. Molecular volumes were predicted with the online tool “molinspiration” [34], using the isomers described in more detail in Appendix A. The nomenclature based on acronyms and the common chemical names was used from previous reports [35]. The following abbreviations were used for the nomenclature of the PET oligomers: C (cyclic structure), L (linear structure), TPA (terephthalic acid), EG (ethylene glycol), DEG (diethylene glycol).

### 2.2. Dependence of the Migration on the Diffusion Coefficient

The mass transfer (migration) of substances from FCMs into food or drinks depends on various factors: The concentration of the substance in the material (C_P,0_), contact time and temperature, diffusion coefficient (D_P_), partition coefficient (K_P/F_), surface-to-volume ratio, thickness of the material, and the type of material [7,14,36,37]. If all these factors are known, migration into food or drinks can be predicted by use of diffusion modeling. However, D_P_ is usually unknown for oligomers from FCMs. To address this knowledge gap, several diffusion coefficient prediction models have been developed. Most of these prediction models are over-estimative, which means that the predicted diffusion coefficient is higher, resulting in a (significantly) higher predicted migration compared to real migration into food. For compliance evaluation purposes of PET packaging materials, such overestimative approaches are useful [7,10], because the migration of the oligomers is predicted as worst-case. However, for exposure evaluation and risk assessment, realistic diffusion coefficients should be available. So far, validated parameters that allow realistic diffusion modeling for PET are available for one model [9,14]. The main difference to other models is that this prediction model is based on experimentally determined activation energies of diffusion E_A_ [14], whereas the conventional prediction model is based on a fixed activation energy of 100 kJ/mol for all migrants independent from their molecular weight or volume. Consideration of activation energies is important because activation energies describe the effect of temperature on diffusion coefficients D_P_. This is important if elevated temperatures are applied, as it is the case for PET microwave and ovenable trays.

Comparison of experimentally reported and predicted diffusion coefficients helps to determine differences and applicability of D_P_ predictions and their models for PET oligomers. Since only diffusion coefficients at high temperatures and for the first series cyclic trimer C[TPA+EG]3 were available in the literature [5], the comparisons were limited to this oligomer and reported conditions. Diffusion coefficients were predicted using the activation energy-based (E_A_-based) model [14] as well as the A_P_ model (realistic case and upper limit) [7,38] (Table 2). The A_P_ parameter describes the basic diffusion behavior of the PET polymer matrix towards the diffusion of migrants. The results show that the reported diffusion coefficients (D_P_) for the cyclic PET trimer C[TPA+EG]3 [5] are in great alignment with the predicted D_P_ from the activation energy E_A_-based and the A_P_ model (realistic case). The A_P_ model (upper limit) is intentionally overestimative, as reflected in the much higher diffusion coefficients compared to the other two models, making it less suitable for realistic migration modeling and thus for this study.

It should be noted, that the diffusion coefficients are determined at very high temperatures of 115–176 °C. The activation energy-based model is validated only for temperature of about 120 °C [9]. Therefore, the comparison of the diffusion coefficients of the cyclic PET trimer at 149 °C and 176 °C is subject to some uncertainty despite their comparability. However, as mentioned above, the activation energy-based prediction model is considering the influence of temperature on the diffusion coefficients on the individual substances. This leads to a more realistic prediction at all temperatures [9] compared to the A_P_ model with a fixed activation energy. The activation energy based prediction model is therefore an applicable method and important to realistically evaluate PET oligomers and their migration potential at different temperatures. Due to the fact, that the A_P_ model is still recommended as standard prediction model for migration prediction [7,37], both prediction models were used in this study.

### 2.3. Dependence of the Migration on the Partition Coefficient

Along with the diffusion coefficient (D_P_), the partition coefficient (K_P/F_) is an additional factor that contributes to the migration modeling results. A major limitation is that for NIAS the partition coefficient between PET and food (simulants) is generally unknown and no practical or validated prediction models for partition coefficients are available. On the other hand, the partition coefficient plays only a role when the equilibrium between the polymer and the contact medium is (nearly) reached [39]. This means that the influence of partitioning is increased with increasing diffusion (e.g., small molecules and/or high temperatures) or the FCM has a (very) low thickness [40]. The impact of partitioning can be assessed and was successfully demonstrated by evaluating the migration under conditions using two different partition coefficients, representing high or low partitioning for simulations [39]. To investigate the contribution of the partition coefficient for PET oligomers, migration of all 52 molecules was simulated for K_P/F_ = 1 and K_P/F_ = 1000 in agreement with the JRC Modeling guidance document [7]. K_P/F_ = 1 can be considered as good solubility of the oligomer in food (worst-case), whereas K_P/F_ = 1000 simulates low solubility of the oligomers in food (best-case). The impact of the partition coefficient on the migration of PET oligomers was calculated for five different applications (time/temperature) with temperatures between 25 °C and 100 °C and application times of 10 min up to 365 days (d). In addition, two different material thicknesses of 10 µm and 2 mm were used to determine whether the effect changes with the thickness and thus the different use of the PET material. The deviations of the results expressed in percent (%) deviation between K_P/F_ = 1 and K_P/F_ = 1000 are summarized in Table 3. The diffusion coefficients used for the migration calculation were predicted either from the E_A_-based model (marked with E_A_) or the A_P_ model (marked with A_P_). The color gradient in Table 3 indicates impact of K_P/F_ with colorless to dark red with increasing impact.

The following conditions were used for the predictions:

Initial concentration in the polymer (C_P,0_) = 1000 mg/kg (has no influence, because the impact is expressed in % as a relative concentration);

Material thickness of 10 µm or 2 mm, which represents the two extremes in PET food contact applications;

Surface-to-volume ratio of 6 dm^2^ per 1 kg food, which is the typical ratio used for compliance evaluation, when no further information is available.

The following application conditions (temperatures and time) were used:

Condition 1: 25 °C for 365 d;

Condition 2: 40 °C for 60 d;

Condition 3: 70 °C for 30 min;

Condition 4: 100 °C for 10 min;

Condition 5: 100 °C for 2 h.

The same results were obtained for the two material thicknesses 10 µm and 2 mm. For the E_A_-based and the A_P_ model, the results for both conditions are shown combined in Table 3. Typically, PET trays or bottles, which are the major applications of PET in the food packaging market, are between these two extremes and have a layer thickness of 200–300 µm. Thus, the material thickness can be neglected, since it has no influence on the migrated amount of the oligomers and is therefore independent of the application conditions used in this study. Considering a variability of ≤10% as acceptable (no dependency on K_P/F_), as this is a common variability of migration in experimental tests [39], no dependency on K_P/F_ for the E_A_-based model for all oligomers in applications with low temperatures ≤ 25 °C, even under long-term storage conditions of up to 365 days was observed. The dependency of K_P/F_ is identical for applications at 100 °C (10 min and 2 h), reaching up to 16% deviation for the smallest molecules (<205 Å^3^, 236 g/mol). When diffusion coefficients were predicted with the A_P_ model, the influence of the partition coefficients is generally higher. This is due to the fact that for the A_P_ model the diffusion coefficients are significantly higher compared to the E_A_-based model. When using diffusion coefficients from the A_P_ model, K_P/F_ has an impact of up to 28% on the predicted migration already for conditions at 25 °C (365 d) and 40 °C (60 d) for oligomers ≤ 342 Å^3^ (402 g/mol) and exhibits an even stronger deviation of up to 40% at 100 °C (10 min and 2 h) for molecules ≤ 427 Å^3^ (491 g/mol).

For the interpretation of the results it is important to notice, that typical storage conditions of PET packed food is at ambient temperature. In addition, many of the oligomers are high molecular weight substances. Therefore, for conditions at ambient or elevated temperatures the partition coefficient is negligible in the migration calculations for PET (E_A_-based model) and should to be considered for oligomers ≤ 342 Å^3^ (402 g/mol) for the A_P_ model. In case of heating applications, the partition coefficient should be taken into account for oligomers < 205 Å^3^ (236 g/mol) for the E_A_-based model and ≤ 427 Å^3^ (491 g/mol) for the A_P_ model. However, as mentioned above, the partition is in most cases unknown. Therefore, two calculations should be made, one with a low (K_P/F_ = 1, worst-case) and one with a high partition coefficient (K_P/F_ = 1000, best-case) in order to estimate the concentration range of the predicted migration.

### 2.4. Migration Modeling of PET Oligomers

To evaluate the migration potential of PET oligomers, 12 different storage or application scenarios were used for predictions. These conditions include applications such as storage at ambient or elevated temperatures (25 °C or 40 °C, respectively) for beverage bottles and short-term or long-term heating applications (70 °C or 100 °C) for microwaveable or ovenable trays (see Materials and Methods for details). A maximum application temperature of 100 °C was selected, based on typical application conditions for PET and assuming that the water content in food dominates and therefore most foods will not reach temperatures higher than 100 °C [41]. In addition to the application conditions, D_P_, K_P/F_, and the initial concentrations of migrants within the polymer (C_P,0_) are needed for migration predictions and were collected from the literature for all PET oligomers [5,13,18,31,32,35,42,43,44,45,46,47,48,49]. For 29 out of 52 oligomers data on C_P,0_ was available. For four oligomers, the distribution of diethylene glycol units was not defined and therefore the C_P,0_ values could not be assigned to a single isomer. For all other oligomers, the C_P,0_ values correspond to the molecules and their respective SMILES as defined in Appendix A. Where multiple concentration values were reported for an oligomer, the values were combined into a range for each literature source (Appendix A).

To predict migration within a realistic range and compare results, the lowest and highest C_P,0_ value (C_P,0_ min and C_P,0_ max, respectively) was used to predict the migration for each oligomer. Identical C_P,0_ max and C_P,0_ min values were used for oligomers with only one C_P,0_ value (Appendix A). Migration predictions were performed with all 12 application conditions using the E_A_-based model as well as the A_P_ model (realistic case). For all molecules for which a dependency on partitioning was identified (deviation of >10%, Table 3) migration predictions were performed using both partition coefficients K_P,F_ = 1 and K_P,F_ = 1000. For all other oligomers only one partition coefficient, K_P/F_ = 1, was used for the predictions. The obtained migration values were then set into a chemical risk assessment context and the migration was compared to thresholds usually applied for potential DNA-reactive mutagens/carcinogens (0.0025 µg/kg body weight/day), Cramer Class I (30 µg/kg body weight/day), and Cramer Class III (1.5 µg/kg body weight/day) substances. A conventional assumption for evaluating migration and exposure levels in adults is the daily consumption of 1 kg of food by a 60 kg person and that the food is packaged in a cubic container with a surface area of 6 dm^2^ [6]. Therefore, migration thresholds corresponding to daily exposure levels for adults can be set at 1.8 mg/kg food for linear PET oligomers, categorized as Cramer Class I substances (if no DNA-reactive mutagens/carcinogens), 90 µg/kg food for cyclic oligomers, categorized as Cramer Class III substances (if no DNA-reactive mutagens/carcinogens), and 0.15 µg/kg food for DNA-reactive mutagens/carcinogens. The results are graphically summarized in Figure 1 and Appendix A for the E_A_-based model and in Figure 2 and Appendix A for the A_P_ model. Predicted values for all oligomers applying all conditions and parameters are available in the Appendix A.

As expected, migration decreases with decreasing concentrations of the migrant in the polymer, resulting in lower migration predictions for C_P,0_ min compared to C_P,0_ max. Additionally, there is a general reduction in migration for the food contact articles in the order 500 mL bottle > 1000 mL bottle > 1500 mL bottle and for rectangular tray > round tray, which is due to the reduction in the surface-to-volume ratios in the same order. Migration is a process that depends on temperature as well as time [7], and a decrease in migration can also be observed following the order: (100 °C, 2 h) > (100 °C, 10 min) > (40 °C, 60 days) > (70 °C, 30 min) > (25 °C, 365 days). Comparing the predictions of the E_A_-based model with those of the A_P_ model, migration is higher for all conditions due to the consideration of higher diffusion coefficients in the A_P_ model. Despite this difference, both prediction models show similar trends. For neither of the models, the linear oligomers exceed the Cramer Class I threshold under all applied conditions and both C_P,0_ concentrations. Additionally, no Cramer Class III threshold violation was observed for cyclic oligomers when C_P,0_ min values were used. Only one oligomer, the cyclic trimer C[TPA+EG]3 for the E_A_-based model (Figure 1b) and two oligomers, the first series cyclic trimer C[TPA+EG]3 and first series cyclic tetramer C[TPA+EG]4 for the A_P_ model (Figure 2b), exceed the Cramer Class III threshold by applying C_P,0_ max values. The main difference between the models is that for the E_A_-based model this can only be observed for a single heating condition (rectangular tray, 100 °C, 2 h) (Figure 1b). In the A_P_ model, C[TPA+EG]3 exceeds the Cramer Class III threshold already at 25 °C (365 d) for all bottle sizes and the C[TPA+EG]4 for the 500 mL bottle (Appendix A). The cyclic trimer exceeds for the rectangular tray at 70 °C (30 min) (Appendix A) and for both trays at 100 °C (10 min and 2 h) the Cramer Class III threshold (Figure 2b and Appendix A). C[TPA+EG]4 violates the Cramer Class III threshold at 100 °C (2 h) for the rectangular tray (Figure 2b). In case of oligomers, for which migration was also predicted with K_P/F_ = 1000, the partition coefficient had no influence on the interpretation of the result, i.e., whether a threshold was violated or not, and can therefore be neglected. Considering the Cramer I and III safety thresholds to be applicable, these results show that for almost all oligomers and conditions used in this study, no significant risk to the consumer can be expected. On the other hand, the first series cyclic trimer has been regularly detected at high concentrations in migration studies and was even detected in human blood [50]. Therefore, a more thorough evaluation is strongly recommended for C[TPA+EG]3 to determine whether the elevated concentration in food may pose a risk to consumers.

Besides the comparison of predicted migrations to Cramer Class I and III thresholds, an additional evaluation was conducted using the threshold for DNA-reactive mutagens and carcinogens. Given the very low allowable concentration (0.15 µg/kg) of such substances, exceedances of the threshold were observed for almost all oligomers under heating conditions and for both prediction models (Figure 1a and Figure 2a). However, when applying the E_A_-based model for conditions at 25 °C up to 70 °C, the majority of oligomers do not exceed the safety threshold for DNA-reactive mutagens/carcinogens (Figure 1a). For the A_P_ model, in contrast, the results at 25 °C are comparable to those at 100 °C and migration levels are even higher at 25 °C for 365 d (Appendix A and Figure 2a). The minor difference between those conditions is due the fixed activation energies in the A_P_ model, not correctly considering the temperature influence on the diffusion coefficient D_P_. When comparing the impact of the partition coefficient K_P,F_ on the migration predictions between the two models, K_P/F_ is again neglectable for the E_A_-based model, but has to be considered for the A_P_ model when interpreting the results. In case of, for example, the first series cyclic monomer C[TPA+EG] and using C_P,0_ min for predictions, the migration would be above the threshold with K_P/F_ = 1, but below the threshold for K_P/F_ = 1000. These results show that for exposure assessments it is important to be aware of the differences between the available prediction models for diffusion coefficients as well as the impact of the different parameters and conditions on the migration process.

As almost all oligomers migrate under several conditions at levels above the safety threshold of DNA-reactive mutagens/carcinogens, it is important to further evaluate these substances by in silico and/or in vitro genotoxicity testing. This has already been partially studied in an initial in silico assessment, in which many oligomers, also included in the present study, were investigated using various available genotoxicity assessment tools. None of the oligomers evaluated was found to be of concern due to potential genotoxicity [26]. Given the structural similarity of the oligomers examined in the present study, sharing the same functional groups and differing essentially only in size, it can be assumed that this can be applied to the entire group of oligomers. However, for confirmation purposes, an additional in vitro evaluation with a selection of representative oligomers should be considered.

It is important to mention that the comparison with the safety thresholds has been performed on the basis of reported concentrations of the oligomers in the literature [5,13,18,31,32,35,42,43,44,45,46,47,48,49]. These C_P,0_ values are therefore snapshots of the PET materials investigated in these previously reported studies and might not be generally applicable for all PET packaging materials. In addition, the oligomeric content of PET FCMs might be influenced by the process conditions and recyclate content. In a circular study on PET bottles, a loop of 11 recycling cycles has been performed [51]. The concentrations of seven oligomers have been monitored during the recycling cycles, and as a general trend the concentrations of the oligomers were decreasing with increasing numbers of recycling cycles. Due to the increasing use of recyclates in all kind of PET packaging materials, the concentration of the PET oligomers might decrease, which has a positive influence on the safety evaluation conducted in this section.

### 2.5. Modeling of C_P,0_ for PET Oligomers

As mentioned above, the concentrations of PET oligomers are often unknown for each PET packaging on the market. In addition, (future) trends like the increasing use of recyclates in PET packaging might change the concentrations of PET oligomers in FCMs. Therefore, the safety evaluation in the previous section is a snapshot for the concentrations of the oligomers reported in the literature (Appendix A). Because of this uncertainty, another method of evaluation, a reversed migration approach, was used in this study. Based on the same contact conditions and predicted diffusion coefficients, the concentration (C_P,0_) in the packaging material was calculated, which corresponds to the migration limits for all three conditions using thresholds for Cramer Class I and III substances, and DNA-reactive mutagens/carcinogens (1.8 mg/kg, 90 µg/kg, and 0.15 µg/kg, respectively). This theoretical C_P,0_ can be considered as the maximum concentration in the PET without violation of the above-mentioned migration threshold limits. All calculations were performed using migration predictions with K_P/F_ = 1 and therefore to evaluate the worst-case and thus the minimum tray or bottle wall concentrations (C_P,0_) required to reach these thresholds.

The obtained results reflect the trends observed and discussed for migration predictions. With decreasing surface-to-volume and increasing molecular volume, an increase in the bottle or tray wall concentration is required to achieve migrations in the range of the respective safety threshold. Additionally, tray or bottle wall concentrations have to be generally higher in the E_A_-based model compared to the A_P_ model due to the differences in diffusion coefficients D_P_ (Figure 3). To assess the probability and associated risk of reaching the individual thresholds, a value of 1% (w/w) was assumed as a realistic and applicable limit. Concentrations above 1% are considered unlikely in this study, since even for the most abundant as well as tested first series cyclic trimer, concentrations > 1% were not observed (Appendix A). A limit of 1% also results in high confidence in the results and interpretation, since for concentrations up to 1% the validity of the generally accepted physical law of diffusion is given [7].

Therefore, a threshold concentration of 1% was used for the evaluation of the calculated C_P,0_, with concentrations below this limit being more likely to be observed in PET articles than concentrations above it. The obtained results showed that for both models a migration level of 0.15 µg/kg can be reached with C_P,0_ < 1% for almost all oligomers and used conditions. Some exceptions are molecules with high M_V_, ≥849 Å^3^ for conditions at 25 °C (365 d), ≥1289 Å^3^ at 40 °C (60 d), and ≥1104 Å^3^ at 70 °C (30 min, only round tray) when using the E_A_-based model (Figure 3a,c). However, as discussed above, none of the previously analyzed oligomers showed in silico results of concern, so the application of the threshold for DNA-reactive mutagens and carcinogens is most certainly not a requirement to ensure consumer safety. However, based on the publicly available data, this cannot be categorically excluded at this stage.

The C_P,0_ concentrations required to reach migrations of 90 µg/kg (Cramer Class III) are significantly higher for all conditions (Figure 3b,d). As a result, when the E_A_-based model is applied, only very small oligomers ≤ 205 Å^3^ at 25 °C (365 d), ≤223 Å^3^ at 40 °C (60 d), and ≤384 Å^3^ at 70 °C (30 min) or 100 °C (10 min) and also for oligomers with higher M_V_ of ≤646 Å^3^ at 100 °C (2 h) reach this threshold with C_P,0_ concentration below 1%. In contrast, for almost all oligomers (≤1146 Å^3^) concentrations below 1% are sufficient to reach migration levels of 90 µg/kg when applying the A_P_ model for conditions at ambient or elevated temperatures. For short-term heating applications (70 °C, 30 min and 100 °C, 10 min), oligomers ≤ 892 Å^3^ reach the threshold with C_P,0_ concentrations below 1%, while for long-term heating applications (100 °C, 2 h), this is the case for even almost all oligomers (≤1146 Å^3^). At this point it must be mentioned that a calculated C_P,0_ of >1 × 10^6^ mg/kg corresponds to a content of > 100% of the oligomer in the polymer, which is of course not possible and reasonable. However, these values were not removed because they reflect clearly the impossibility of exceeding or even reaching the respective safety threshold, as they would require concentrations that simply cannot exist. In such cases it can be clearly ruled out that such conditions will ever pose a safety risk. This becomes even more clear for concentrations needed to reach migrations of 1.8 mg/kg (Cramer Class I) (Appendix A). At ambient and elevated temperatures, all oligomers require concentrations of >1% for both models and even almost all >100% with the E_A_-based model. This results are also comparable for all heating applications and all oligomers ≥ 266 Å^3^, with one exception for oligomers ≤ 427 Å^3^ at 100 °C for 2 h and the rectangular tray (A_P_ model). Reaching migration levels of 1.8 mg/kg is therefore very unlikely, and linear oligomers can be considered safe for most applications. However, evaluation of small oligomer under especially long-term heating applications is still recommended.

This results show that even with the lack of C_P,0_ values and exposure evaluation can be performed in a systematic way identifying conditions and/or oligomers of more or less concern. This can very helpful for prioritization purposes in the risk assessment context. Another advantage of this approach is that these maximum concentrations of the oligomers in PET can be used as reliable quality parameters for monitoring production [36].

## 3. Materials and Methods

### 3.1. Diffusion Modelling

AKTS SML software version 4.54 and 6.41 (AKTS AG, Siders, Switzerland) was used for the modeling of all migrations. The model is based on a Finite Element Analysis (FEA) and a description of the diffusion model is published [52]. The molinspiration online tool [34] was used for the calculation of molecular volumes (M_V_) for all PET oligomers. For the activation energy-based E_A_ model the validated parameters for the prediction of the diffusion coefficient were: a = 1.93 × 10^−3^ 1/K, b = 2.37 cm^2^/s, c = 11.1 Å^3^ and d = 1.50 × 10^−5^ 1/K [9,14]. For the prediction of diffusion coefficients with the A_P_ model the following parameter were used: A’_P_ = 3.1 and τ = 1577 (realistic case) or A’*_P_ = 6.4 and τ = 1577 (upper limit) [7,10,38].

### 3.2. Properties and Parameters Used for Migration Modeling

Properties and parameters applied for the predictions were used according to Table 4, Appendix A, unless otherwise stated. Storage/application conditions for beverage bottles were used and adjusted based on the exposure study conducted in Munich, Germany, in 2007 [40] and the guideline for testing of kitchenware articles [53]:

End of shelf-life under controlled storage conditions: 365 days, 25 °C;

Short-term non-controlled/extreme summer storage condition: 60 days, 40 °C;

Short-term heating in microwave or oven: 10 min, 100 °C or 30 min, 70 °C;

Long-term heating in oven: 2 h, 100 °C.

The temperature of 100 °C was chosen under assumption that the water content in food is dominating and therefore most food will reach not temperatures higher than 100 °C [41].

## 4. Conclusions

Experimental migration tests on PET oligomers are time consuming and the results are associated with considerable uncertainty, mainly due to the lack of reference standards for most oligomers. The low concentrations in the migration solutions as well as potential swelling effects or hydrolysis when using aqueous ethanolic solutions as food simulants, make the interpretation of the results very difficult.

The use of migration models represents a useful tool for the safety evaluation of PET oligomers. However, diffusion coefficients need to be available for all PET oligomers. Experimentally determined diffusion coefficients are rare in the scientific literature and only available at very high temperatures for the PET first series cyclic trimer. Therefore, prediction models for diffusion coefficients D_P_ are necessary for the migration modeling approach. The results for the diffusion coefficients depend on the prediction model for the diffusion coefficients, based on the over-estimative character of the applied prediction model.

Small oligomers with high diffusion coefficients are the most critical substances that can also pose the greatest risk to consumers, especially when the storage conditions include high temperatures, e.g., in microwave or ovenable trays. In these cases, the prediction model for the diffusion coefficients should include the influence of temperature on the diffusion coefficients. This makes the E_A_-based model more suitable for a realistic evaluation of the consumer exposure. In addition, the E_A_-based prediction model is validated for many different substances and temperatures, resulting in a more realistic modeling approach and therefore a more reliable exposure prediction of oligomers. However, the over-estimative A_P_ prediction model can be also applied, resulting in a more conservative evaluation of the results, when typical concentrations available from the literature are applied.

The second major parameter influencing the migration process, the partition coefficient K_P,F_, plays a minor role and can be neglected in most cases. This is due to the slow diffusion of high molecular weight compounds like PET oligomers in PET. If the partition coefficient can be neglected, the procedure is simplified, because K_P,F_ = 1 can be used for worst-case prediction. In case of doubts, e.g., for small molecules and/or high temperatures a second modeling test should be performed with K_P,F_ = 1000 in order to investigate the influence on partitioning on the predicted migration results.

The third important parameter in the exposure evaluation of PET is the concentration of oligomers in PET (C_P,0_). These values are often not available from the scientific literature or might change in the future due to increase recyclate content or optimized production conditions. Calculation of the maximum concentration of oligomers in PET solves this problem, and as a positive side effect, reliable quality control parameters are available for production control. The evaluation of C_P,0_ required to reach safety threshold can be a very useful way of exposure assessment by helping to identify conditions that are more or less likely to be reached and therefore associated with higher or lower risk for the consumer, respectively. This approach can also be readily applied to evaluate the relevance of concentrations associated with observed adverse effects or for other populations with lower safety limits. Additionally, it can further be used for other migrants in PET as PET oligomers.

As an overall conclusion of this study, migration modeling represents a very helpful tool for a systematic exposure evaluation and prioritization of oligomers and/or conditions of concern. It provides a fast, comparable, and comprehensive overview that can easily be used for risk assessment purposes on the migration properties of PET oligomers besides the lack of migration testing data. In addition to the generally known evidence that heating applications and small molecules are of greater concern than low temperature conditions and large oligomers, this study could provide specific numbers needed for a comprehensive exposure assessment for this particular group of substances.

## Figures and Tables

**Figure 1 molecules-28-00173-f001:**
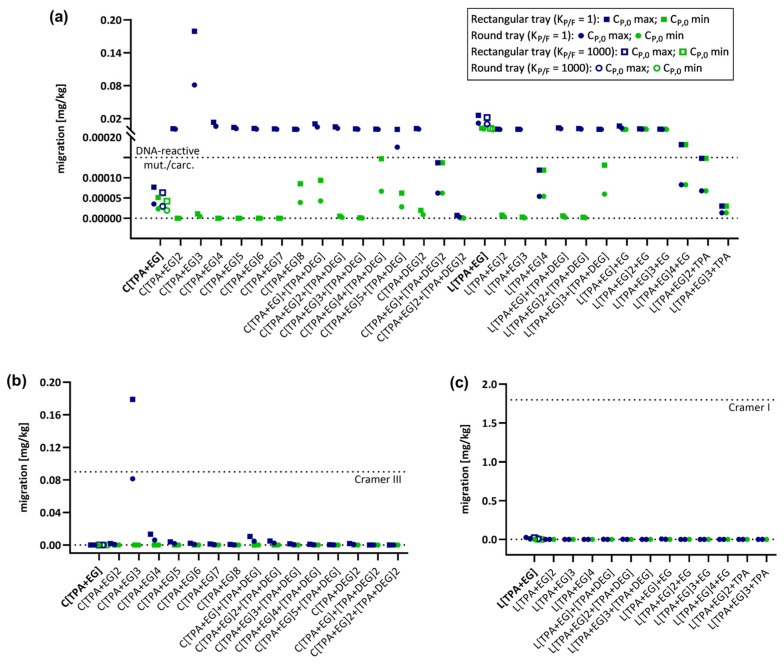
Graphical representation of predicted migrations using the activation energy E_A_-based model. Migration of PET oligomers for which C_P,0_ values were available was predicted for PET trays (rectangular and round) under long-term heating conditions (100 °C, 2 h) and evaluated for the exceeding different safety thresholds. Oligomer names in bold represent molecules for which both K_P/F_ = 1 and K_P/F_ = 1000 were used for predictions. (**a**) Modeled migration of all 29 PET oligomers and their evaluation against the DNA-reactive mutagens/carcinogens safety threshold of 0.15 µg/kg. (**b**) Modeled migration of cyclic PET oligomers and their evaluation against the Cramer Class III threshold of 90 µg/kg. (**c**) Modeled migration of linear PET oligomers and their evaluation against the Cramer Class I threshold of 1.8 mg/kg.

**Figure 2 molecules-28-00173-f002:**
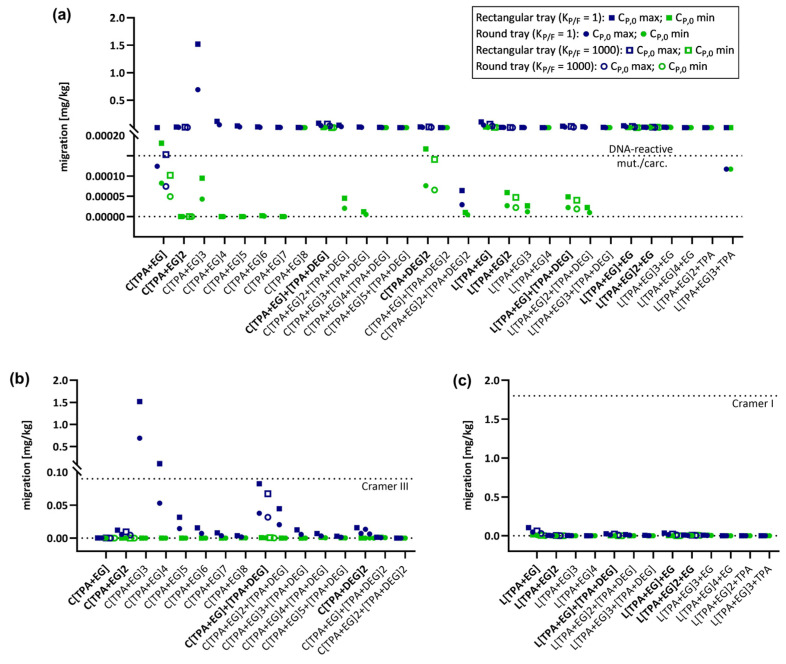
Graphical representation of predicted migrations using the A_P_ model (realistic case). Migration of PET oligomers for which C_P,0_ values were available was predicted for PET trays under long-term heating conditions (100 °C, 2 h) and evaluated for the exceeding different safety thresholds. Oligomer names in bold represent molecules for which both K_P/F_ = 1 and K_P/F_ = 1000 were used for predictions. (**a**) Modeled migration of all 29 PET oligomers and their evaluation against the DNA-reactive mutagens/carcinogens safety threshold of 0.15 µg/kg. (**b**) Modeled migration of cyclic PET oligomers and their evaluation against the Cramer Class III threshold of 90 µg/kg. (**c**) Modeled migration of linear PET oligomers and their evaluation against the Cramer Class I threshold of 1.8 mg/kg.

**Figure 3 molecules-28-00173-f003:**
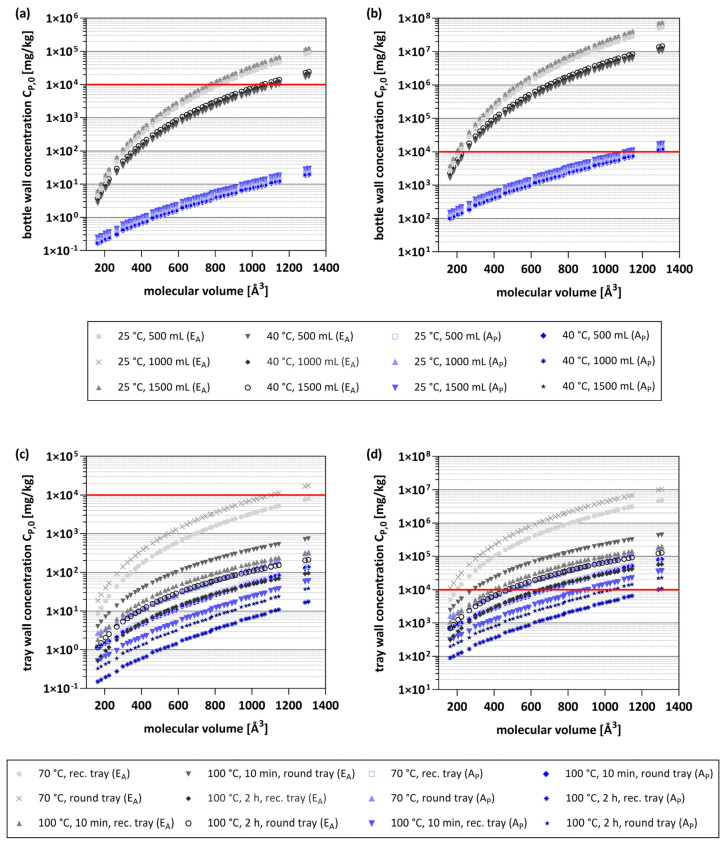
Maximum bottle and tray wall concentrations (C_P,0_) of all oligomers corresponding to a migration of 90 µg/kg or 0.15 µg/kg, calculated for the E_A_-based model (grey) and the A_P_ model (blue). (**a**) C_P,0_ concentrations for 500, 1000, and 1500 mL bottle at 25 °C for 365 d and 40 °C for 60 d corresponding to a migration of 0.15 µg/kg. (**b**) C_P,0_ concentrations for 500, 1000, and 1500 mL bottles at 25 °C for 365 d and 40 °C for 60 d corresponding to a migration of 90 µg/kg. (**c**) C_P,0_ concentrations for rectangular (rec.) and round trays at 70 °C for 30 min and 100 °C for 10 min or 2 h corresponding to a migration of 0.15 µg/kg. (**d**) C_P,0_ concentrations for rectangular (rec.) and round trays at 70 °C for 30 min and 100 °C for 10 min and 2 h corresponding to a migration of 90 µg/kg. Red line indicates a C_P,0_ concentration of 1% (w/w).

**Table 1 molecules-28-00173-t001:** Summary of reported PET oligomers and hypothetical molecules. PET oligomer names (acronyms) of the hypothetical PET oligomers are highlighted in grey.

PET Oligomer (Acronym)	Common Oligomer Name	Molecular Weight M_W_ [g/mol]	Predicted Molecular Volume M_V_ [Å^3^]
C[TPA+EG]	First series cyclic monomer	192.17	162.74
C[TPA+EG]2	First series cyclic dimer	384.34	323.69
C[TPA+EG]3	First series cyclic trimer	576.51	484.64
C[TPA+EG]4	First series cyclic tetramer	768.68	645.59
C[TPA+EG]5	First series cyclic pentamer	960.85	806.54
C[TPA+EG]6	First series cyclic hexamer	1153.02	967.49
C[TPA+EG]7	First series cyclic heptamer	1345.19	1128.43
C[TPA+EG]8	First series cyclic octamer	1537.36	1289.38
C[TPA+DEG]	Second series cyclic monomer	236.22	205.33
C[TPA+EG]+[TPA+DEG]	Second series cyclic dimer	428.39	366.28
C[TPA+EG]2+[TPA+DEG]	Second series cyclic trimer	620.56	527.23
C[TPA+EG]3+[TPA+DEG]	Second series cyclic tetramer	812.73	688.18
C[TPA+EG]4+[TPA+DEG]	Second series cyclic pentamer	1004.9	849.12
C[TPA+EG]5+[TPA+DEG]	Second series cyclic hexamer	1197.07	1010.07
C[TPA+DEG]2	Third series cyclic dimer	472.45	408.87
C[TPA+EG]+[TPA+DEG]2	Third series cyclic trimer	664.62	569.82
C[TPA+EG]2+[TPA+DEG]2	Third series cyclic tetramer	856.79	730.76
C[TPA+EG]3+[TPA+DEG]2	Third series cyclic pentamer	1048.96	891.71
C[TPA+EG]4+[TPA+DEG]2	Third series cyclic hexamer	1241.13	1052.66
C[TPA+EG]+[TPA+DEG]3	Fourth series cyclic tetramer	900.84	773.35
L[TPA+EG]	First series linear monomer	210.19	180.63
L[TPA+EG]2	First series linear dimer	402.36	341.58
L[TPA+EG]3	First series linear trimer	594.52	502.53
L[TPA+EG]4	First series linear tetramer	786.70	663.48
L[TPA+EG]5	First series linear pentamer	978.87	824.43
L[TPA+EG]6	First series linear hexamer	1171.04	985.38
L[TPA+EG]7	First series linear heptamer	1363.20	1146.33
L[TPA+EG]8	First series linear octamer	1555.38	1307.28
L[TPA+DEG]	Second series linear monomer	254.24	223.22
L[TPA+DEG]+EG	Second series linear monomer + EG	298.29	265.81
L[TPA+EG]+[TPA+DEG]	Second series linear dimer	446.41	384.17
L[TPA+EG]2+[TPA+DEG]	Second series linear trimer	638.58	545.12
L[TPA+EG]3+[TPA+DEG]	Second series linear tetramer	830.75	706.07
L[TPA+EG]4+[TPA+DEG]	Second series linear pentamer	1022.92	867.02
L[TPA+EG]5+[TPA+DEG]	Second series linear hexamer	1215.09	1027.97
L[TPA+DEG]2	Third series linear dimer	490.46	426.76
L[TPA+EG]+[TPA+DEG]2	Third series linear trimer	682.63	587.71
L[TPA+EG]2+[TPA+DEG]2	Third series linear tetramer	874.80	748.66
L[TPA+EG]3+[TPA+DEG]2	Third series linear pentamer	1066.97	909.61
L[TPA+EG]4+[TPA+DEG]2	Third series linear hexamer	1259.14	1070.55
L[TPA+EG]+EG	First series linear monomer + EG	254.24	223.22
L[TPA+EG]2+EG	First series linear dimer + EG	446.41	384.17
L[TPA+EG]3+EG	First series linear trimer + EG	638.58	545.12
L[TPA+EG]4+EG	First series linear tetramer + EG	830.75	706.07
L[TPA+EG]5+EG	First series linear pentamer + EG	1022.92	867.02
L[TPA+EG]6+EG	First series linear hexamer + EG	1215.09	1027.97
L[TPA+EG]+TPA	First series linear monomer + TPA	358.30	299.00
L[TPA+EG]2+TPA	First series linear dimer + TPA	550.47	459.94
L[TPA+EG]3+TPA	First series linear trimer + TPA	742.64	620.89
L[TPA+EG]4+TPA	First series linear tetramer + TPA	934.81	781.84
L[TPA+EG]5+TPA	First series linear pentamer + TPA	1126.98	942.79
L[TPA+EG]6+TPA	First series linear hexamer + TPA	1319.15	1103.74

**Table 2 molecules-28-00173-t002:** Comparison of reported experimentally determined diffusion coefficients (D_P_) for C[TPA+EG]3 [5] and predicted values from the E_A_-based prediction model [14] and the A_P_ model (realistic case and upper limit) [7,38].

Temperature [°C]	D_P_ Measured [cm^2^/s]	D_P_ Predicted [cm^2^/s]		
		E_A_ Based Model	A_P_ Model (Realistic Case)	A_P_ Model (Upper Limit)
176	2.9 × 10^−9^	1.4 × 10^−9^	2.5 × 10^−10^	6.9 × 10^−9^
149	6.6 × 10^−10^	3.8 × 10^−11^	4.6 × 10^−11^	1.2 × 10^−9^
115	1.2 × 10^−12^	2.0 × 10^−13^	3.8 × 10^−12^	1.0 × 10^−10^

**Table 3 molecules-28-00173-t003:** Impact of partitioning at different conditions for PET oligomers predicted by the E_A_-based model or the A_P_ model (realistic case). For each condition, the deviation of migration in mg/dm^2^ between the partition coefficients K_P/F_ = 1 and K_P/F_ = 1000 is given in %. The color gradient indicates the impact of K_P/F_, with the influence increasing as the intensity of red increases.

PET Oligomer (Acronym)	M_W_ [g/mol]	M_V_ [Å^3^]	Impact of the Partition Coefficient (K_P/F_ = 1 and K_P/F_ = 1000) on Migration in % Condition/Model *									
			1/E_A_	2/E_A_	3/E_A_	4/E_A_	5/E_A_	1/A_P_	2/A_P_	3/A_P_	4/A_P_	5/A_P_
C[TPA+EG]	192.17	162.74	1	2	2	16	16	26	28	14	40	40
L[TPA+EG]	210.19	180.63	1	1	2	13	13	24	25	13	37	37
C[TPA+DEG]	236.22	205.33	0	1	1	9	10	21	23	11	34	34
L[TPA+DEG]	254.24	223.22	0	1	1	8	8	20	21	10	31	31
L[TPA+EG]+EG	254.24	223.22	0	1	1	8	8	20	21	10	31	31
L[TPA+DEG]+EG	298.29	265.81	0	0	0	5	5	16	17	8	27	27
L[TPA+EG]+TPA	358.3	299.00	0	0	0	4	4	12	13	6	21	21
C[TPA+EG]2	384.34	323.69	0	0	0	3	3	11	12	5	19	19
L[TPA+EG]2	402.36	341.58	0	0	0	3	3	10	11	5	18	18
C[TPA+EG]+[TPA+DEG]	428.39	366.28	0	0	0	2	2	9	10	4	16	16
L[TPA+EG]+[TPA+DEG]	446.41	384.17	0	0	0	2	2	9	9	4	15	15
L[TPA+EG]2+EG	446.41	384.17	0	0	0	2	2	9	9	4	15	15
C[TPA+DEG]2	472.45	408.87	0	0	0	2	2	8	8	4	14	14
L[TPA+DEG]2	490.46	426.76	0	0	0	2	2	7	8	3	13	13
L[TPA+EG]2+TPA	550.47	459.94	0	0	0	1	1	6	6	3	10	10
C[TPA+EG]3	576.51	484.64	0	0	0	1	1	5	6	2	9	9
L[TPA+EG]3	594.52	502.53	0	0	0	1	1	5	5	2	9	9
C[TPA+EG]2+[TPA+DEG]	620.56	527.23	0	0	0	1	1	4	5	2	8	8
L[TPA+EG]2+[TPA+DEG]	638.58	545.12	0	0	0	1	1	4	4	2	8	8
L[TPA+EG]3+EG	638.58	545.12	0	0	0	1	1	4	4	2	8	8
C[TPA+EG]+[TPA+DEG]2	664.62	569.82	0	0	0	1	1	4	4	2	7	7
L[TPA+EG]+[TPA+DEG]2	682.63	587.71	0	0	0	1	1	3	4	2	7	7
L[TPA+EG]3+TPA	742.64	620.89	0	0	0	1	1	3	3	1	5	5
C[TPA+EG]4	768.68	645.59	0	0	0	1	1	3	3	1	5	5
L[TPA+EG]4	786.7	663.48	0	0	0	1	1	2	3	1	5	5
C[TPA+EG]3+[TPA+DEG]	812.73	688.18	0	0	0	1	1	2	2	1	4	4
L[TPA+EG]3+[TPA+DEG]	830.75	706.07	0	0	0	0	0	2	2	1	4	4
L[TPA+EG]4+EG	830.75	706.07	0	0	0	0	0	2	2	1	4	4
C[TPA+EG]2+[TPA+DEG]2	856.79	730.76	0	0	0	0	0	2	2	1	4	4
L[TPA+EG]2+[TPA+DEG]2	874.8	748.66	0	0	0	0	0	2	2	1	3	4
C[TPA+EG]+[TPA+DEG]3	900.84	773.35	0	0	0	0	0	2	2	1	3	3
L[TPA+EG]4+TPA	934.81	781.84	0	0	0	0	0	2	2	1	3	3
C[TPA+EG]5	960.85	806.54	0	0	0	0	0	1	2	1	3	3
L[TPA+EG]5	978.87	824.43	0	0	0	0	0	1	1	1	3	3
C[TPA+EG]4+[TPA+DEG]	1004.9	849.12	0	0	0	0	0	1	1	1	2	2
L[TPA+EG]4+[TPA+DEG]	1022.92	867.02	0	0	0	0	0	1	1	1	2	2
L[TPA+EG]5+EG	1022.92	867.02	0	0	0	0	0	1	1	1	2	2
C[TPA+EG]3+[TPA+DEG]2	1048.96	891.71	0	0	0	0	0	1	1	1	2	2
L[TPA+EG]3+[TPA+DEG]2	1066.97	909.61	0	0	0	0	0	1	1	1	2	2
L[TPA+EG]5+TPA	1126.98	942.79	0	0	0	0	0	1	1	0	2	2
C[TPA+EG]6	1153.02	967.49	0	0	0	0	0	1	1	0	2	2
L[TPA+EG]6	1171.04	985.38	0	0	0	0	0	1	1	0	1	1
C[TPA+EG]5+[TPA+DEG]	1197.07	1010.07	0	0	0	0	0	1	1	0	1	1
L[TPA+EG]5+[TPA+DEG]	1215.09	1027.97	0	0	0	0	0	1	1	0	1	1
L[TPA+EG]6+EG	1215.09	1027.97	0	0	0	0	0	1	1	0	1	1
C[TPA+EG]4+[TPA+DEG]2	1241.13	1052.66	0	0	0	0	0	1	1	0	1	1
L[TPA+EG]4+[TPA+DEG]2	1259.14	1070.55	0	0	0	0	0	1	1	0	1	1
L[TPA+EG]6+TPA	1319.15	1103.74	0	0	0	0	0	1	1	0	1	1
C[TPA+EG]7	1345.19	1128.43	0	0	0	0	0	0	1	0	1	1
L[TPA+EG]7	1363.2	1146.33	0	0	0	0	0	0	0	0	1	1
C[TPA+EG]8	1537.36	1289.38	0	0	0	0	0	0	0	0	1	1
L[TPA+EG]8	1555.38	1307.28	0	0	0	0	0	0	0	0	1	1

* For storage conditions 1 to 5, see text. Abbreviations: E_A_ (E_A_-based prediction model), A_P_ (A_P_ model, realistic case), M_W_ (molecular weight), M_V_ (molecular volume)

**Table 4 molecules-28-00173-t004:** Properties and parameters used for migration modeling for different food contact articles.

Properties and Conditions	Food Contact Articles				
	500 mL Bottle	1.0 L Bottle	1.5 L Bottle	Rectangular Tray *	Round Tray **
Contact surface [cm^2^]	420	660	880	566	329
Contact volume [L]	0.5	1.0	1.5	0.5	637
Surface/volume [1/cm^2^]	0.84	0.66	0.59	1.13	0.52
Thickness PET [µm]	300	300	300	300	300
Density PET [g/cm^3^]	1.4	1.4	1.4	1.4	1.4
Density food/drink [g/cm^3^]	1.0	1.0	1.0	1.0	1.0
K_P/F_	1 or 1000	1 or 1000	1 or 1000	1 or 1000	1 or 1000
Temperature and time(application scenarios)	**Condition 1**	**Condition 1**	**Condition 1**	**Condition 3**	**Condition 3**
	25 °C, 365 d	25 °C, 365 d	25 °C, 365 d	70 °C, 0.5 h	70 °C, 0.5 h
	**Condition 2**	**Condition 2**	**Condition 2**	**Condition 4**	**Condition 4**
	40 °C, 60 d	40 °C, 60 d	40 °C, 60 d	100 °C, 10 min	100 °C, 10 min
				**Condition 5**	**Condition 5**
				100 °C, 2 h	100 °C, 2 h

* Trays with the assumed surface in contact (500 mL filling): h = 3.25 cm, w = 12 cm, l = 16 cm. ** Trays with the assumed surface in contact (637 mL filling): h = 4.8 cm, r = 6.5 cm.

## Data Availability

The data presented in this study are available on request from the corresponding author.

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
