# Peer review of "Migration Modeling as a Valuable Tool for Exposure Assessment and Risk Characterization of Polyethylene Terephthalate Oligomers"

_molecules, 2022, doi:10.3390/molecules28010173_

Round 1

Reviewer 1 Report

Dear authors,

In my opinion, the objective of the study is clear and very relevant for migration modeling.

The manuscript is introduced with a clear problem setting and need for the study. Though I do miss the relevance of the study for recycled PET in the introduction. 

The applied methods seem realistic and the test conditions have been chosen to be representative for the actual food contact material domain of PET. The authors build further on their previous published work.

The data is clearly explained and the relevance of the results is linked to realistic/practical cases of PET bottles and trays.

Some suggestions:

Lines 90-91: the formulation of the last part of the sentence "... the potential exposures for consumers associated with individual oligomers and/or applications", could be improved.

Line 129: "The concentration of the migration in the material ..." : formulation should be checked.

Line 133: this gap or these gaps?

Line 215-216: Can you better explain how table 3 shows that there is no difference in material thickness (10 µm vs 2 mm)? It's not clear for me from lines 195-196.

Line 288: There is a copy of figure 1 between line 288 and 289?

Line 388: oligomer(s)

Best regards.

Author Response

Dear Reviewer, thanks for your valuable review. We considered all of your recommendations.

In my opinion, the objective of the study is clear and very relevant for migration modeling.

The manuscript is introduced with a clear problem setting and need for the study. Though I do miss the relevance of the study for recycled PET in the introduction.

The applied methods seem realistic and the test conditions have been chosen to be representative for the actual food contact material domain of PET. The authors build further on their previous published work.

The data is clearly explained and the relevance of the results is linked to realistic/practical cases of PET bottles and trays.

Answer: Thank you very much for taking the time to review our manuscript. We appreciate your positive feedback and are glad to hear that you find the objective of the study to be clear and relevant for migration modeling. We have taken your suggestion to include the relevance of the study for recycled PET in the introduction and have made this addition. We have carefully considered the points you raised and made the necessary revisions to the manuscript. The responses to each comment are detailed below. Thank you for your valuable feedback and look forward to the opportunity to resubmit the revised manuscript for your further consideration.

Regarding not mentioning PET recyclates in the introduction. rPET is indeed an important point, but the topic of oligomers and the migration from PET is not a recyclate specific topic. Therefore we focused the introduction on PET and not on rPET. The oligomers are also not increasing during recycling, which had been shown in Lit 51.

Some suggestions:

Lines 90-91: the formulation of the last part of the sentence "... the potential exposures for consumers associated with individual oligomers and/or applications", could be improved.

Line 129: "The concentration of the migration in the material ..." : formulation should be checked.

Line 133: this gap or these gaps?

Line 388: oligomer(s)

Answer: Thank you for bringing this to our attention. We have made changes to the formulation of the text in lines 90-91, 129, 133, and 388 to address your comments and hope that the revised version meets your approval.

Lines 90-91: “In order to understand the potential migration of PET oligomers under realistic conditions of use, migration modeling was employed in this study. The goal of this evaluation was to determine the potential level of consumer exposure that may be linked to specific oligomers and/or applications. Through this work, we aimed to show the usefulness of migration modeling for PET oligomers and its ability to support the compliance and risk assessment process for this group of substances.”

Lines 129: “The mass transfer (migration) of substances from FCMs into food or drinks depends on various factors: The concentration of the substance in the material (CP,0), contact time and temperature, diffusion coefficient (DP), partition coefficient (KP/F), surface-to-volume ratio, thickness of the material, and the type of material.”

Line 133: “this gap”

Lines 388: “oligomers”

Line 288: There is a copy of figure 1 between line 288 and 289?

Answer: This was a wrong link. We deleted the additional figure.

Line 215-216: Can you better explain how table 3 shows that there is no difference in material thickness (10 µm vs 2 mm)? It's not clear for me from lines 195-196.

Answer: Thank you for your suggestion. We have taken it into consideration and revised the formulation of the sentences (lines 215-216) to clearly indicate that the results presented in Table 3 are combined for both material thicknesses: “The same results were obtained for the two material thicknesses 10 µm and 2 mm. For the EA-based and the AP model, the results for both conditions are shown combined in Table 3. “ We hope this addresses your concern.

Reviewer 2 Report

Manuscript ID: molecules-2087139

The authors report on the migration of a variety of 52 oligomer substances from polyethylene terephthalate (PET) using the activation energy-based (EA) model and the AP model.

The use of migration modeling to assess the migration potential of PET oligomers and characterize their risk after degradation is an intriguing approach. Recently, there have been few reports on the fate of the thermoplastic polymers after degradation.

I think that the work should be published with regard to a well-thought-out plan and its undoubted interest in practice. However, the presentation of the results needs improvement in accordance with the following comments, which should be addressed prior to publication. Therefore, I am suggesting minor revisions. 

1.      Table 1: What do the highlights in the PET oligomer (acronym) column mean? Please explain in detail in the footnote.

2.      Table 3: The authors should mention the meaning of the highlight in the column of the impact of the partition coefficient (KP/F = 1 and KP/F = 1000) on migration in % Condition/Model*. In my opinion, the table should be self-explanatory for the readers. Please indicate the meaning of the red highlights in the footnote. 

Author Response

Dear Reviewer, thanks for your valuable review. We considered all of your recommendations.

The authors report on the migration of a variety of 52 oligomer substances from polyethylene terephthalate (PET) using the activation energy-based (EA) model and the AP model.

The use of migration modeling to assess the migration potential of PET oligomers and characterize their risk after degradation is an intriguing approach. Recently, there have been few reports on the fate of the thermoplastic polymers after degradation.

Answer: Thank you very much for taking the time to review our manuscript. We are glad that you find the use of migration modeling to assess the migration potential of PET oligomers and characterize their risk after degradation to be an intriguing approach. We appreciate your recognition of the relevance and interest of our work in practice. We have carefully considered the points you raised and made the necessary revisions to the manuscript. The responses to each comment are detailed below. Thank you for your valuable feedback and look forward to the opportunity to resubmit the revised manuscript for your further consideration.

I think that the work should be published with regard to a well-thought-out plan and its undoubted interest in practice. However, the presentation of the results needs improvement in accordance with the following comments, which should be addressed prior to publication. Therefore, I am suggesting minor revisions.

  1. Table 1: What do the highlights in the PET oligomer (acronym) column mean? Please explain in detail in the footnote.

Answer: Thank you for bringing this to our attention. We have rephrased the text in the table heading of Table 1 to make it clearer: “PET oligomer names (acronyms) of the hypothetical PET oligomers are highlighted in grey.“

  1. Table 3: The authors should mention the meaning of the highlight in the column of the impact of the partition coefficient (KP/F = 1 and KP/F = 1000) on migration in % Condition/Model*. In my opinion, the table should be self-explanatory for the readers. Please indicate the meaning of the red highlights in the footnote.

Answer: Thank you for bringing this to our attention as well. We have now added a sentence to clarify the meaning of the red highlight in the table heading of Table 3: “The color gradient indicates the impact of KP/F, with the influence increasing as the intensity of red increases.” We hope this addresses your concern.

Reviewer 3 Report

The manuscript “Migration modelling as a valuable tool for exposure assessment and risk characterization of polyethylene terephthalate oligomers” constitutes an interesting study, well written and organised and can be considered for publication in “Molecules” after considering the following points.

1.- Please, define the meaning of the abbreviation “Ap” model in line 18.

2.- Authors mention (lines 52-54) that mixtures of PET with aqueous ethanol can swell the polymer causing over-estimation of migration, and readers can interpret it as a disadvantage in comparison with modelling methods. This is not necessarily negative, since it corresponds to a worst case, and if real migration values are below the limits, that means additional safety. In fact, the Ap model also overestimates most of times the expected or measured migration values and it is considered as acceptable or desirable (lines 137-138 & 518-520).

3.- Conditions 1-5 (lines 210-214) do not correspond exactly with the ones equally designed in table 4, which can be confusing. In fact, in such table, conditions 2, 4 and 6 seem to be the same, and the same for conditions 7 &10, 8 & 11 and 9 & 12. Please, detail it better or simplify.

4.- “Chapter” (lines 388 and 395) could be replaced by “section” or “paragraph”

5.- In lines 444-449 the use of “needed” or “required” can be confusing, when speaking about the cases <1%. It could be replaced by “are enough” or “are detected at concentrations below 1%” or similar.

6.- Conclusions should be condensed to emphasize the main findings. In my opinion, now are a (too long) summary of the work.

Author Response

Dear Reviewer, thanks for your valuable review. We considered all of your recommendations, except the recommendation to shorten the conclusions section.

The manuscript “Migration modelling as a valuable tool for exposure assessment and risk characterization of polyethylene terephthalate oligomers” constitutes an interesting study, well written and organised and can be considered for publication in “Molecules” after considering the following points.

Answer: Thank you very much for taking the time to review our manuscript. We are grateful for your positive feedback and appreciate your thorough review of our work. We have carefully considered the points you raised and made the necessary revisions to the manuscript. Thank you for your valuable feedback and look forward to the opportunity to resubmit the revised manuscript for your further consideration.

1.- Please, define the meaning of the abbreviation “Ap” model in line 18.

Answer: Thank you for bringing this to our attention. The AP parameter is mentioned as abbreviation in the Lit 5 and in the JRC Modeling guidance document (Lit 7). We added the sentence "The AP parameter describes the basic diffusion behavior of the PET polymer matrix towards the diffusion of migrants."

2.- Authors mention (lines 52-54) that mixtures of PET with aqueous ethanol can swell the polymer causing over-estimation of migration, and readers can interpret it as a disadvantage in comparison with modelling methods. This is not necessarily negative, since it corresponds to a worst case, and if real migration values are below the limits, that means additional safety. In fact, the Ap model also overestimates most of times the expected or measured migration values and it is considered as acceptable or desirable (lines 137-138 & 518-520).

Answer: We certainly agree with your point and have therefore added the following sentence: “Overestimating migration levels is beneficial for safety and regulatory compliance, but it is less useful for risk and exposure assessment as it can lead to an overestimation of the actual risk.” (lines 55-56)

3.- Conditions 1-5 (lines 210-214) do not correspond exactly with the ones equally designed in table 4, which can be confusing. In fact, in such table, conditions 2, 4 and 6 seem to be the same, and the same for conditions 7 &10, 8 & 11 and 9 & 12. Please, detail it better or simplify.

Answer: Thank you for bringing this issue to our attention. We have now revised Table 4 to ensure that the conditions are consistently described by using “conditions 1-5” throughout the manuscript. We hope that this addresses your concern.

4.- “Chapter” (lines 388 and 395) could be replaced by “section” or “paragraph”

Answer: Thank you. We have replaced "chapter," as you suggested, with "section."

5.- In lines 444-449 the use of “needed” or “required” can be confusing, when speaking about the cases <1%. It could be replaced by “are enough” or “are detected at concentrations below 1%” or similar.

Answer: We have taken your suggestion into consideration and revised the relevant parts of the text accordingly: “The CP,0 concentrations required to reach migrations of 90 µg/kg (Cramer Class III) are significantly higher for all conditions (Figure 3b and 3d). As a result, when the EA-based model is applied, only very small oligomers ≤ 205 Å3 at 25 °C (365 d), ≤ 223 Å3 at 40 °C (60 d), and ≤ 384 Å3 at 70 °C (30 min) or 100 °C (10 min) and also for oligomers with higher MV of ≤ 646 Å3 at 100 °C (2 h) reach this threshold with CP,0 concentration below 1%. In contrast, for almost all oligomers (≤ 1146 Å3) concentrations below 1% are sufficient to reach migration levels of 90 µg/kg when applying the AP model for conditions at ambient or elevated temperatures. For short-term heating applications (70 °C, 30 min and 100 °C, 10 min), oligomers ≤ 892 Å3 reach the threshold with CP,0 concentrations below 1%, while for long-term heating applications (100 °C, 2 h), this is the case for even almost all oligomers (≤ 1146 Å3).” (lines 445-455).

6.- Conclusions should be condensed to emphasize the main findings. In my opinion, now are a (too long) summary of the work.

Answer: We discussed this recommendation in the group of authors and we came to the conclusion, that in our point of view the conclusions are not too long. The manuscript has 19 pages and the conclusions section is about 1 page. Therefore, we decide not to shorten the conclusion section. Hope for your understanding.